# Crustacean Waste-Derived Chitosan: Antioxidant Properties and Future Perspective

**DOI:** 10.3390/antiox10020228

**Published:** 2021-02-03

**Authors:** Manikandan Muthu, Judy Gopal, Sechul Chun, Anna Jacintha Prameela Devadoss, Nazim Hasan, Iyyakkannu Sivanesan

**Affiliations:** 1Laboratory of Neo Natural Farming, Chunnampet, Tamil Nadu 603 401, India; bhagatmani@gmail.com; 2Department of Environmental Health Sciences, Konkuk University, Seoul 05029, Korea; jejudy777@gmail.com (J.G.); scchun@konkuk.ac.kr (S.C.); 3New Prince Shri Bhavani Senior Secondary School, Ullagaram, Chennai 600 091, India; awesomeanna@email.com; 4Department of Chemistry, Faculty of Science, Jazan University, Jazan P.O. Box 114, Saudi Arabia; nazim7862000@gmail.com; 5Department of Bioresources and Food Science, Institute of Natural Science and Agriculture, Konkuk University, 1 Hwayang-dong, Gwangjin-gu, Seoul 05029, Korea

**Keywords:** chitosan, chitin, crustacean shells, waste, antioxidant, derivatives, applications, nanotechnology

## Abstract

Chitosan is obtained from chitin that in turn is recovered from marine crustacean wastes. The recovery methods and their varying types and the advantages of the recovery methods are briefly discussed. The bioactive properties of chitosan, which emphasize the unequivocal deliverables contained by this biopolymer, have been concisely presented. The variations of chitosan and its derivatives and their unique properties are discussed. The antioxidant properties of chitosan have been presented and the need for more work targeted towards harnessing the antioxidant property of chitosan has been emphasized. Some portions of the crustacean waste are being converted to chitosan; the possibility that all of the waste can be used for harnessing this versatile multifaceted product chitosan is projected in this review. The future of chitosan recovery from marine crustacean wastes and the need to improve in this area of research, through the inclusion of nanotechnological inputs have been listed under future perspective.

## 1. Introduction

Chitin is the second most abundant (first is cellulose), non-toxic, biodegradable high molecular weight polymer occurring in nature. It exhibits outstanding chemical and biological properties, such as biocompatibility, non-toxicity, biodegradability, and exceptional adsorption properties. These unique properties have enabled chitin to be used in industrial and biomedical applications. Chitin is the major component of cuticles of insects (cockroach, beetle, true fly, and worm), fungal cell walls (*Aspergillus niger*, *Mucor rouxii*, *Penicillum notatum*, yeast) and green algae [1,2,3,4]. Chitin is a constituent of different exoskeletons of marine arthropods such as crustaceans (crab, shrimp, lobster, krill, crayfish, barnacles), cuttlefish, and squid pen. Chitin is made up of a linear chain of acetylglucosamine groups [5]. With a turnover of 10 billion tons annually [6,7], chitin is one of the most abundant biopolymers. Chitin can be readily obtained by simple extraction [8] and the major source of industrial chitin is derived from wastes of marine food production, mainly crustacean shells, e.g., shrimp, crab, or krill shells [9,10,11]. Crustacean shell is composed of 30–40% proteins, 30–50% mineral salts, and 13–42% of chitin occurring in α-, β-, and γ-forms. In the processing of shrimps for human consumption, between 40 and 50% of the total mass is waste and 40% of this waste is chitin. A small part of the waste is usually dried and utilized as chicken feed [11], while the rest is dumped into the sea, which is one of the main pollutants in coastal areas [12,13]. The utilization of shellfish waste has thus been able to solve environmental problems, being an alternative waste disposal method [14,15]. In practice, a polymer where most residues are acetylated is chitin, while the opposite holds well for chitosan. Several processes have been proposed for the use of shellfish waste for extraction of chitin and chitosan. Chitin extraction involves alkaline extraction of organics and acid solubilization/decomposition of minerals, subsequent to protein removal. During such extractions, the chitin molecule also suffers some structural changes, including a moderate degree of deacetylation. Chitosan, is obtained on substantial deacetylation through alkaline treatment of chitin under severe conditions.

Research in the area of renewable marine byproducts has recently been patronized. The current application of crustacean waste mainly focuses on the value-added production of chitin. Chemical methods of chitin extraction use voluminous hazardous chemicals (NaOH and HCl) that are released into the environment. The abundant and renewable marine processing wastes are commercially exploited for the extraction of chitin. However, the traditional chitin extraction processes employ harsh chemicals at elevated temperatures for a prolonged time which can harm chitin’s physico-chemical properties and are also held responsible for the deterioration of environmental health. There are mainly two chitin extraction methods conducted in the industry, those being chemical or biological. Both extraction strategies of chitin consist of two phases, deproteinization with alkaline treatment at high temperatures followed by demineralization predominantly with dilute hydrochloric acid. The sequence of these two phases is interchangeable depending on the source and the proposed use of chitin. During the chemical process, the protein component in the shells is not recovered, however, this is resolved through enzymatic processing methods resulting in the effective recovery of proteins, chitin, and pigments [16,17]. Chitin powder isolated from crustacean sources has a pale pink color, thus necessitating the bleaching process involving hydrogen peroxide, oxalic acid, or potassium permanganate [18,19]. In order to avoid the use of harsh chemicals, green extraction methods are increasingly gaining popularity due to their environmentally friendly nature. Chitin and its derivatives are widely used in innumerable applications ranging from food, agriculture, biomedicine, pharmaceuticals, and cosmetics to environmental sector. Figure 1 gives an overview of the overall process involved in the recovery of chitosan from crustacean shell wastes.

Biotechnological production of chitin offers new perspectives for the production of highly viscous chitosan, with promising inputs in biomedicine and pharmacy [20,21]. Proteolytic enzymes such as, papain, alkalase, chymotrypsintrypsin, pepsin, devolvase and pancreatin have been employed for the extraction and separation of chitin and proteins from shrimp waste [22,23]. Mhamdi et al. reported the use of serine alkaline proteases from *Micromonospora chaiyaphumensis* S103 for chitin extraction from shrimp shell (*Penaeus kerathurus*) waste [24]. Other authors have described that the application of crude digestive alkaline proteases from *Portunus segnis* led to effective extraction of chitin by deproteinization of blue crab (*P. segnis*) and shrimp (*P. kerathurus*) [25]. The most used bacterial strains for fermentation are *Lactobacillus* sp. strain especially *L. plantarum*, *L. paracasei*, and *L. helveticus*. Recently, Castro et al. extracted and purified chitin from *Allopetrolisthes punctatus* crabs using *Lactobacillus plantarum* sp.47, a Gram-positive bacterium isolated from Coho salmon that produces high lactic acid concentrations [26]. For chitin recovery with non-lactic acid bacteria, crustacean shell fermentation bacteria and fungi such as *Pseudomonas* sp., *Bacillus* sp. and *Aspergillus* sp. were used as the inoculum source: Ghorbel-Bellaaj et al. isolated a protease bacterium identified as *Pseudomonas aeruginosa* A2. Most commercial bacterial proteases are mainly produced by *Bacillus* sp., and Hajji et al. have extracted from the waste of crab shells chitin and fermented-crab supernatants after fermentation using six different strains of *Bacillus*. The bioextraction of chitin from crustacean shell wastes has been increasingly researched at the laboratory scale, but not at the commercial level. Bioextraction of chitin is thus a greener, cleaner, eco-friendly and economical process. Specifically, microorganisms-mediated fermentation processes are highly desirable due to easy handling, simplicity, rapidity, controllability through optimization of process parameters, ambient temperature, and negligible solvent consumption, thus reducing environmental impact and costs. Driven by reduced energy, wastewater, or solvent, advances in biological extraction of chitin along with valuable byproducts will have high economic and environmental impact.

However, chitosan properties and applications and recovery methods have been already reviewed. The following review highlights the recovery of chitosan from crustacean wastes and emphasizes on the need to seek out “green” extraction methods. The available data on the antioxidant properties of chitosan have been reviewed and the need to expand the research focus in this area of research for harnessing the full potential of the antioxidant properties of chitosan has been proposed. Nanotechnological inputs into areas of chitosan recovery and furtherance of biological activity (especially antioxidant activity) of chitosan have been highlighted as a necessary improvement.

## 2. Chitin–Chitosan Metamorphosis

The seafood processing industry produces large quantities of byproducts and discards such as heads, tails, skins, scales, viscera, backbones, and shells of marine organisms. Although these are waste residues, they still are an excellent source of lipids, proteins, pigments, and small molecules, and moreso a source of chitinous materials. One of the limitations in the use of chitin on a large-scale is its water insolubility, this is why water-soluble derivatives have been sought after. Chitosan is the most important of these. Chitosan is obtained from chitin by a process called deacetylation, whereby acetyl groups (CH_3_-CO) are removed resulting in a molecule that is soluble in most diluted acids [27]. The deacetylation process releases amine groups (-NH_2_) rendering chitosan a cationic nature. Chitosan, a linear polysaccharide is made up of β-(1–4)-linked d-glucosamine and N-acetyl-d-glucosamine moieties [28,29,30,31,32,33,34]. Chitosan is derived from chitin by chemical or enzymatic deacetylations. Although chemical deacetylation is preferentially cheaper and warrants mass production, the major disadvantage is the energy consumption and high environmental pollution risks. Alternatives in the form of enzymatic methods that utilize chitin deacetylases have been explored via enzymatic deacetylation of chitin. Research has identified that selected fungal, viral, and bacterial chitin deacetylases could produce partially acetylated chitosan tetramers with a defined degree of acetylation and a pattern of acetylation [35]. With the recent progress in extraction methodologies and instrumentation sophistication, chitosan extraction from marine crustaceans has also been achieved outside of chemical extraction through autoclave-based methods [36,37,38,39,40,41,42], microwave-based methods [39,43,44,45,46], ultrasonication-based methods [47,48,49], and Graviola extract combined with magnetic stirring on hot plate [50] (Table 1). Table 1 tabulates the published chitosan recovery methods specifically from crustacean wastes, the techniques employed and the recovery variables have been reported. To date, the available recovery options include chemical-, autoclave-, ultrasonication-based methods, and a phytoextract-based methodology using graviola leaf extracts. The predominant recovery method demonstrated is chemical recovery and the least reported is phytoextract-mediated green recovery. This review urges the need to focus research in the area of green extraction.

Chitin and chitosan exhibit several biological properties such as anti-cancer [109], antioxidant [110], antimicrobial [111], and anti-coagulant [112] properties. In addition, they are used as biomaterials in a wide range of applications: for biomedical purposes such as for artificial skin, bones, and cartilage regeneration [113,114], for food preservation such as for edible films [115], and for pharmaceutical purposes such as for drug delivery [116]. Chitin is a versatile, environmentally friendly, modern material [117]. Chitin and chitin derivatives have been used in virtually every significant segment of the economy (e.g., water treatment, pulp and paper industry, biomedical devices and therapies, cosmetics, biotechnology, agriculture, food science, and membrane technology) [118]. Chitin and chitosan are important bioactive materials, with many highly potent activities such as immune function, hemostasis and wound healing, antioxidant action, antimicrobial activity, and heavy metal and other pollutant removal [119]. Therefore, as renewable resources, chitin and its derivatives have a wide range of applications in food and nutrition [120], pharmaceutical [121], biotechnological [122], cosmetic [123], packaging [124], textile, wastewater treatment [125], and agricultural [126] industries.

## 3. Snap Shot of the Bioactive Properties of Chitosan

Chitosan has three reactive groups, the primary amine group and the primary and secondary hydroxyl groups at C-2, C-3, and C-6 positions, respectively [127]. Among the three groups, the primary amine at the C-2 position is reported to be responsible for the bioactivity of chitosan [128]. The chemical modification of chitosan adds unique functional properties for use towards biological and biomedical applications [129,130,131,132,133,134,135,136,137,138,139,140,141,142,143,144,145,146,147,148,149,150,151,152,153,154,155,156,157,158,159,160,161]. The biodegradability, biocompatibility, mucoadhesion, hemostatic, analgesic, adsorption enhancer, antimicrobial, anticholesterolemic, and antioxidant attributes of chitosan are those which make it suitable for biomedical applications.

Chitosan has been well established as an alternative to antibiotics, undertaking antimicrobial and antifungal roles. Because of its cationic properties, chitosan is able to lead to membrane-disrupting effects [162,163,164,165], which are higher against Gram-positives than Gram-negatives [165]. The antibacterial activity of chitosan is influenced by the molecular weight of chitosan and allied physicochemical properties. A number of chitosan derivatives with different modifications have shown improved antibacterial activity; in this way, chitosan micro/nanoparticles display unique physical and chemical features [166]. The chitosan nanoparticles penetrate inside the cell, interacting with phosphorus- and sulfur-containing compounds such as DNA and protein, eventually causing damage to the cells [163,164]. Successful experiments were performed using chitosan and reticulated chitosan microparticles against aquaculture related trouble-makers: *Lactococcus garvieae* (Gram +), *Vibrio parahaemolyticus*, and *Vibrio alginolyticus* (Gram −). These microorganisms are the most predominant bacterial pathogens of mariculture industry and are responsible for crucial economic losses in cultured fish and seafood worldwide [167]. The antimicrobial activity of chitin, chitosan, and their derivatives against different groups of microorganisms, such as bacteria, yeast, and fungi, has received considerable attention in recent years [120,168,169].

Traditional chemotherapeutic agents kill actively dividing cells, characteristic of most cancer cells. Cytotoxic drugs continue to play a crucial role in cancer therapy, although side effects such as the destruction of lymphoid and bone marrow cells is inevitable. In this direction, constant efforts to improve cancer therapy-based side effects are sought after. This is why biocompatible anticancer drugs are needed for cancer therapy. The introduction of several groups into chitosan modifies its structure significantly, thereby increasing the biological activity of chitosan. The introduction of sulfates and phenyl groups in carboxymethyl benzylamide dextrans into chitosan, lead to enhanced anticancer activity in breast cancer cells. Sulfated chitosan (SCS) and sulfated benzaldehyde chitosan (SBCS) significantly inhibited cell proliferation, induced apoptosis, and blocked the fibroblast growth factors (FGF)-2-induced phosphorylation of extracellular signal-regulated kinase (ERK) in Middle cranial fossa (MCF)-7 cells [170]. Dialkylaminoalkylation and reductive amination followed by quaternization of chitosan could elicit inhibitory effects on the proliferation of tumor cell lines [171].

Anti-inflammatory refers to the property of reducing inflammation. Theophylline is a drug that reduces the inflammatory effects of allergic asthma but is difficult to administer at an appropriate dosage without causing adverse side effects. The adsorption of theophylline to chitosan nanoparticles modified by the addition of thiol groups has been reported to enhance theophylline absorption by the bronchial epithelium and thereby enhance its anti-inflammatory effects [172]. The beneficial contributions of chitosan and its oligosaccharides include anti-tumor [173], neuroprotective [174], antifungal and antibacterial [175,176], and anti-inflammatory [177] effects. Tissue engineering is applying a combination of cells, engineering and materials methods, and suitable biochemical and physiochemical factors to improve or replace biological functions. Natural and synthetic materials have been used. Chitosan derivatives have been reported for the preparation of several tissue engineering organs, such as skin, bone, liver, nerve, and blood vessels. The chemical modification of chitosan has a profound impact with respect to delivery of different kinds of drugs to a specific place [178]. Chitosan nanoparticles have led to a significant increase in loading capacity and efficacy. The phenolic and polyphenolic compounds with antioxidant effects have been condensed with chitosan to form mutual prodrugs [179,180]. Chitin and chitosan derivatives are ideal candidates as drug carriers in cancer chemotherapy [181]. Apart from these, chitosan has been used in the pharmaceutical industry in drug delivery systems, such as tablets, microspheres, micelles, vaccines, nucleic acids (NAs), hydrogels, and nanoparticles. Chitosan has also been used for pharmaceutical applications, wound healing, and tissue regeneration [182,183]. In tableting, chitosan has been reported as an excipient to delay the release of the active ingredient from the tablets. Chitosan with high molecular weight is more viscous and is used for sustained release of drugs, in order to prolong drug activity, improve therapeutic efficiency, and reduce side effects in oral tablets [183]. Chitosan has also been reported for its use as coating material in drug delivery applications because of its good film-forming and mucoadhesive properties, leading to controlled release of drugs.

## 4. Antioxidant Activity of Chitosan

Being extracted from crustacean’s exoskeleton and fungi cell walls, chitosan products are biocompatible and biodegradable, and their range of applications include food, wastewater treatment, cell culture, cosmetics, textiles, agrochemicals, and medical devices [30]. Additionally, chitosan also exhibits antioxidant activity [184,185] and can be used as a replacement for synthetic antioxidants such as butylated hydroxytoluene (BHT), butylated hydroxy-anisole (BHA), propyl gallate, and tert-butylhydroquinone (TBHQ) [186]. Reactive oxygen species (ROS) such as H_2_O_2_, hydroxyl radicals, and superoxides lead to oxidative stress which is the key behind a wide range of pathologies: cancer [187], cardiovascular disease [188,189], premature aging [190], rheumatoid arthritis, and inflammation [191]. Chitin, similar to vitamin C, exhibits antioxidant effects [192] and can be used as an ingredient for the production of functional foods in order to circumvent age-related and diet-related diseases [193]. Due to oxidation of lipids in food, off-flavors and rancidity manifests, this is why BHT and BHA (synthetic antioxidants) are used. BHT and BHA are well known for their potential health hazards [194], and hence safe and natural alternatives are being sought [195]. The addition of 1% chitosan resulted in 70% decrease in the 2-thiobarbituric acid reactive substance (TBARS) values of frozen meat. Chitosan addition is reported to have lead to chelation of the free iron in heme proteins of meat that are released during processing [196]. In seafoods, oxidation of high concentrations of prooxidants such as hemoglobin and metal ions in fish muscles is also reported [197]. Antioxidant effect of chitosan is reported [198] to be directly proportional to its molecular weight, concentration, and viscosity. Chitosans from crab shell wastes were tested on herring flesh and chitosan with different viscosity were also tested on fish samples. The highest activity was observed with low viscosity chitosan (14 cP) and its action was similar to that of BHA, BHT, and TBHQ. Chitosans are speculated to slowdown lipid oxidation by chelating ferrous ions in fish. This eliminates the prooxidant activity of ferrous ions by preventing their conversion to ferric ion [197]. Kim and Thomas [199] have also reported identical inferences in Atlantic salmon (*Salmo salar*). 

Free radical reaction is connected with several specific human diseases and has gained paramount attention. In the human body, reactive oxygen species (ROS) produced during metabolic process oxidize lipids, proteins, carbohydrates and nucleic material, resulting in oxidative stress [192]. The ROS generated, may activate enzymes that eventually manifest as life-threatening disorders such as cancer, aging, cardiovascular diseases wrinkle formation, rheumatoid arthritis, inflammation, hypertension, dyslipidemia, atherosclerosis, myocardial infraction, angina pectoris, heart failure, and neurodegenerative diseases such as Alzheimer’s, Parkinson’s, and amyotrophic lateral sclerosis [200,201,202,203,204]. The term ROS refers to oxygen-derived free radicals like superoxide, hydroxyl radical, and nitric oxide, and is extended to nonradical oxygen derivatives of high reactivity like singlet oxygen, hydrogen peroxide, peroxynitrite, and hypochlorite [205,206]. Mitochondria in biological cells are responsible for ROS generation [207]. The cellular defense includes enzymes such as catalase, superoxide dismutase, and glutathione peroxidase [200]. When excessive ROS are generated, the defense mechanism is unable to respond appropriately and thus oxidative stress manifests. The antioxidant activity of chitosan has gained paramount importance, with chitosan exhibiting confirmed scavenging activity against various radical species. The degree of deactylation (DDA) and molecular weight (MW) of chitosan determine the scavenging capacity of chitosan [208] and the chitosan NH_2_ groups are responsible for free radical scavenging effect. Mahdy Samar et al. confirmed that a high rate of DDA and low MW of chitosan exhibits higher antioxidant activity [44]. Hajji et al. studied chitosan obtained from Tunisian marine shrimp (*Penaeus kerathurus*) waste (DDA: 88%), crab (*Carcinus mediterraneus*) shells (DDA: 83%), and cuttlefish (*Sepia officinalis*) bones (DDA: 95%) [74] and tested their antioxidant activities. Cuttlefish with 95% DDA exhibited the highest antioxidant activity. Sun et al. [209], reported that chitosan oligomers exhibited stronger scavenging activity with lower MW. Chang et al. [210] demonstrated that lower MW chitosan (~2.2 kDa) greatly impacts the scavenging ability. Although the antioxidant activity of chitosan has been proven, yet it is limited by the lack of an H-atom donor, to serve as a good chain-breaking antioxidant [211]. The scavenging capacity of free radicals is related to the bond dissociation energy of O–H or N–H and the stability of the formed radicals. Due to the presence of strong intramolecular and intermolecular hydrogen bonds in chitosan molecules, the OH and NH_2_ groups find it difficult to dissociate and react with hydroxyl radicals [212]. This is why various modifications of chitosan molecules by grafting functional groups into the molecular structure have evolved. Modification of chitosan by grafting polyphenols, has been observed to enhance the antioxidant activity. This has resulted owing to the synergetic effects obtained from both chitosan and polyphenols [213]. Chito-oligosaccharides (COS) are known to be highly promising compounds for use as natural antioxidants in biological systems [62,214]. Li et al. 2018 [215] have recently investigated the preparation and potential free radical scavenging activity of chitosan derivatives with 1,2,3-triazoles and 1,2,3-triazoliums. Their results indicated that all the chitosan derivatives exhibited higher radical scavenging activity than chitosan and the scavenging ability was further enhanced following the N-methylation of 1,2,3-triazole moieties. Other researchers [216,217] have also reported the antioxidant activity of quaternary ammonium groups in chitosan derivatives.

Antioxidant agents like chitosan play a role in scavenging the free radicals and by inhibiting the oxidative damage caused by free radicals (Figure 2). Antioxidant mechanism of chitosan functions by protecting the host against oxidative stress induced damages via interfering with the oxidation chain reaction. The exact mechanism of free radical scavenging activity of chitosan has still not been established. However, Riaz et al., 2019 [218] put forth a plausible theory that the unsteady free radicals may react with the OH group and the amino group at C-2, C-3, and C-6 positions of the pyranose ring to produce a stable macromolecule. This review calls to attention that this (elucidating the mechanism of antioxidant activity of chitosan) is one of the gray areas worth working on with respect to the antioxidant activity of chitosan.

## 5. The State-of-the-Art Chitosan Trends

With the outputs of nanotechnology having benefitted almost every area of life, inputs from the nano realm in chitosan are no exception. Nanofabrication comprises of top-down approaches and bottom-up approaches that bring about fabrication of nanostructures from native superstructures through successive disintegration or build-up [219]. Chitosan nanofibers and nanowhiskers have been fabricated using methods of disassembly [220,221]. Wet grinding of chitosan flakes, with subsequent high pressure homogenization resulted in 100 to 1000 nm-sized (diameter) chitin nanofibers. Wijesena et al. [222] prepared chitin nanofibrils (diameter of 5 nm) from crab shells applying ultrasonication for mechanical disassembly. Grinding has also been adopted for fabrication of chitin nanofibrils from microfibrils [223]. Chitin nanowhiskers and nanofibers have been prepared using radical assisted oxidation of chitin with 2,2,6,6-tetramethyl piperidine-1-oxyl followed by ultra-sonication [224]. Tanaka et al. [225] reported nanofibrillation of chitin powder under nitrogen gas using ultrasonication. Chitin nanofibers have been reported by top-down approach or by electrospinning [226]. Most of these approaches involve highly acidic or basic solutions, this is why alternative environment-friendly approaches to produce chitin/chitosan nanofibers were sought after. A self-assembly mechanism is one such alternative approach. However, chitosan nanoparticles being insoluble do not self-assemble [227]. Therefore, amphiphilic chitosan derivatives have been constructed for self-assembly [227]. 

Curcumin-encapsulated chitosan nanoparticles have been reported using the sonication method. The results revealed an improved solubility of curcumin with sustained-release pattern [228]. Chitin nanofibers have been extracted from crab shells [229], from prawn shells [230] including shells of Penaeus monodon (black tiger prawn), Marsupenaeus japonicus (Japanese tiger prawn), and Pandaluseous makarov (Alaskan pink shrimp). Their shells were treated using NaOH and HCl solutions to remove proteins and minerals to yield chitin nanofibers with 10–20 nm uniform width and high aspect ratios. Chitin nanofibers have been reported for use as a drug for inflammatory bowel disease. Moreover, chitosan is also employed as a cell culture media due to its biocompatibility and ability to accelerate growth. In many of these studies, chitosan, polylactic acid (PLA) and polyglycolic acid (PGA) are mixed to form films, porous structures, or beads [231].

Several composites have been prepared, such as chitosan with alginate [139], collagen [232], calcium phosphate [233], hydroxyapatite [234], and polysulfone [235]. Chitosan composite materials have been employed in bone tissue engineering. Bone is mostly composed of collagen and hydroxyapatite; researchers have attempted substituting the function of collagen by chitosan [34,236,237]. Various derivatives of chitosan have also been reported, carboxymethyl-chitosan is an amphoteric polymer and the solubility depends on pH. Hydroxypropyl chitosan was prepared from chitosan and propylene epoxide under alkaline conditions. Other chemical modification of chitosan includes esterification of chitosan and N, O-acylation of chitosan using acetyl chloride in MeSO_3_H as solvent [238]. It has been confirmed that acetylation of chitosan substantially improves its antifungal activity [239]. There are few methods to obtain phosphorylated derivatives of chitosan. These chitosan derivatives are necessary owing to their unique biological and chemical properties. They could exhibit bactericidal and osteoinductive properties. Phosphorylated chitosan can be prepared by heating chitosan with orthophosphoric acid in N, N-dimethylformamide (DMF). Chitosan nanoparticles have been prepared and put to use for catalytic functions too [240,241].

Chitosans with degrees of polymerization (DPs) <20 and an average molecular weight less than 3900 Da are called chitooligosaccharides (COS) [242]. This is one of the most trending chitosan derivate types. COS are generated by depolymerization of chitin or chitosan using acid hydrolysis, hydrolysis by physical methods, and enzymatic degradation [243]. Recently, COS has gained paramount attention owing to its pharmaceutical and medicinal applications, due to their nontoxicity and high solubility and positive physiological effects. The health benefits of COS include lowering blood cholesterol, lowering high blood pressure, protective effects against infections, controlling arthritis, improvement of calcium uptake, and enhancing antitumor properties.

## 6. Future Perspective and Conclusions

Disposal in the future will have to cope with more stringent ecological standards, making recovery of valuable byproducts from wastes mandatory. Integral utilization of renewable resources is certainly a goal worth pursuing and can be envisaged as new reclamation technologies are developed. This certainly applies to shellfish processing waste and byproducts. Shellfish waste represents a substantial portion of the raw material. These materials, mainly shells, viscera, heads, and adhered meat, are partly used today to produce fishmeal for animal consumption, the rest being wasted even though it is environmentally offensive. The relative amount of these materials varies with the species and the technology of processing. There is considerable potential for conversion of this waste into value-added products to resolve some of the issues associated with environment pollution and cost of disposal. Chitosan has been extensively expanding and especially the very aspect of recovery of this valuable resource from marine shell waste is by itself a very attractive topic. Given the fact that such a worthwhile recovery on the other hand is also helping through productive use of crustacean wastes, ridding the environment of a bio-pollutant, still the extraction process itself has not been improved much. Only traditional extraction processes, with very minimal use of sophistication in terms of advanced instrumentation have been employed. Green extraction strategies have been minimally attempted and published; more green extraction methods and large scale extraction methods need to be addressed. This is a huge lacuna we observed as this research was reviewed. Muthu et al. have already reported extraction of chitosan from sea waste uisng graviola leaf extracts. This is only the beginning, there are numerous such green aspects that can be probed. This review is expected to draw some response from researchers in this area of research in this specific direction. With chitosan holding huge potentials in industrial, medical, and agricultural applications, there is still need for more chitosan and large-scale recovery units and factories with more advanced technologies. How much crustacean waste are we currently using to extract chitosan? How much more can we utilize this marine waste? Is there a chance that we can perhaps use all of it? Are all questions worth giving a thought or a shot?

Chitosan composite materials with integrated properties are much less reported; this is a crucial area demanding progress. Chitosan derivatives have also not been exploited fully, each derivative has a few reports attached and each derivative has not been extensively assessed of its wholesome properties. COS, which apparently appears to hold a lot of potential, needs to be put forward to various medical applications. Moreover, extraction of COS from crustacean wastes, involving advanced techniques, should be seriously looked into.

Chitosan-based drugs that are employed to treat various cancers are still at the laboratory stages of testing. No clinical trials are reported and most of the medical applications are far from even beginning the journey from benchtop to bedside. This gray area needs to be worked on. There is no doubt that the recovery of chitosan and its derivatives from marine shell wastes is a valuable asset, what remains is how best we can advance the recovery methods and how best we can use the recovered chitosan. Since chitosan has huge potential and given the fact that it can be harnessed from a wasting away resource, everything that it takes to bring chitosan to the forefront needs to be attempted and achieved.

Nanotechnology is a field that has made much room for progress in various areas of research and development. There has been very limited exploitation of this cutting-edge technology in chitosan recovery, in chitosan composites, and in chitosan applications; this is something this review would like to project. Nanotechnology has pushed the limits of various gray areas in various fundamental research, more inclusion of this technology into chitosan research is crucial for progress. When composites have been done with bulk materials such as alginates, collagen, and hydroxyapatites, there is certainly “room for more” making these composites with nano-alginates, nano versions of collagens, and the like. Table 2 presents the nano-chitosan/nano-chitosan composites versions that have been synthesized so far and their deliverables achieved. With such medical applications of nano-chitosan already established [244,245,246,247,248,249,250,251,252,253,254,255,256,257,258,259,260,261,262,263,264,265,266], it is strange that these chitosan nanomaterials have not been assessed for their antioxidant properties. The antioxidant potential of chitosan is far from being harnessed, there has been nothing much reported in the direction of enhancement of the antioxidant potential of chitosan through nanocomposites and a fundamental understanding of antioxidant activity of chitosan (mechanism) is greatly lacking. Moreover, the antioxidant activity of chitosan derivatives has been clearly shown to be higher than that of chitosan, yet other than the existing handful reports there has been low enthusiasm on extending the antioxidant applications to chitosan derivatives.

This review briefly summarized the existing recovery, bioactive, and antioxidant properties and applications of chitosan. The antioxidant property of chitosan has been vaguely and randomly studied and represented by not many authoritative reports, this area needs to be explored more. The most important emphasis here is that since crustacean waste can be turned into a multifaceted beneficial resource; more cutting-edge technologies need to be applied to and improved in chitosan research. This appears to have not been done as we look through what has been achieved, hence we encourage progressive upgradations involving nano aspects and nano-integrated composite approaches as future initiatives.

## Figures and Tables

**Figure 1 antioxidants-10-00228-f001:**
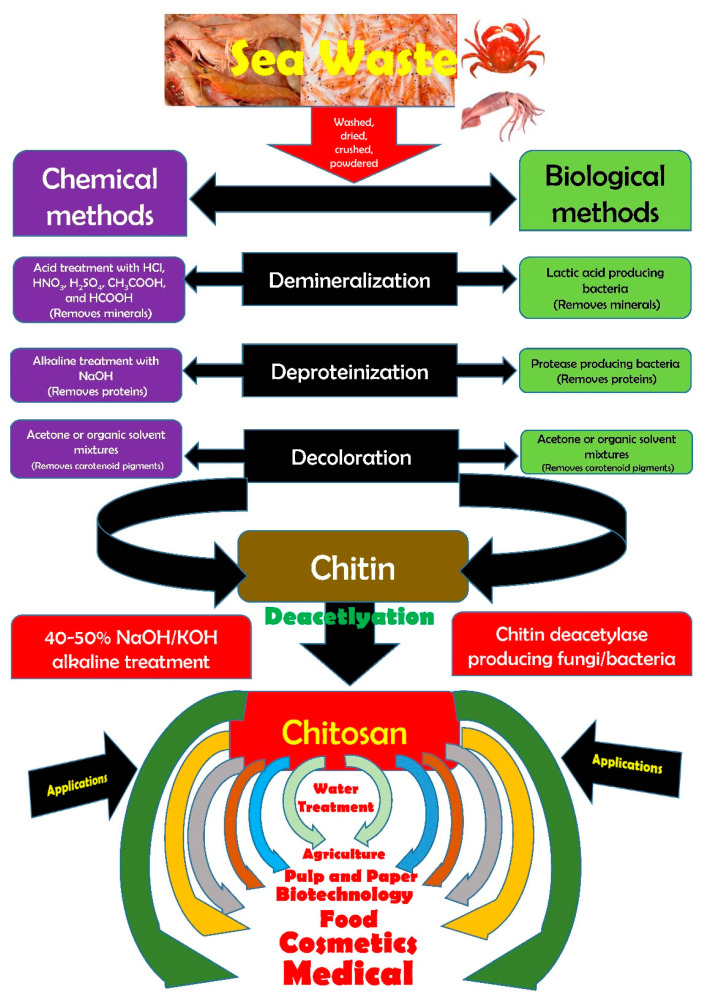
Overview of the recovery of chitosan from crustacean wastes.

**Figure 2 antioxidants-10-00228-f002:**
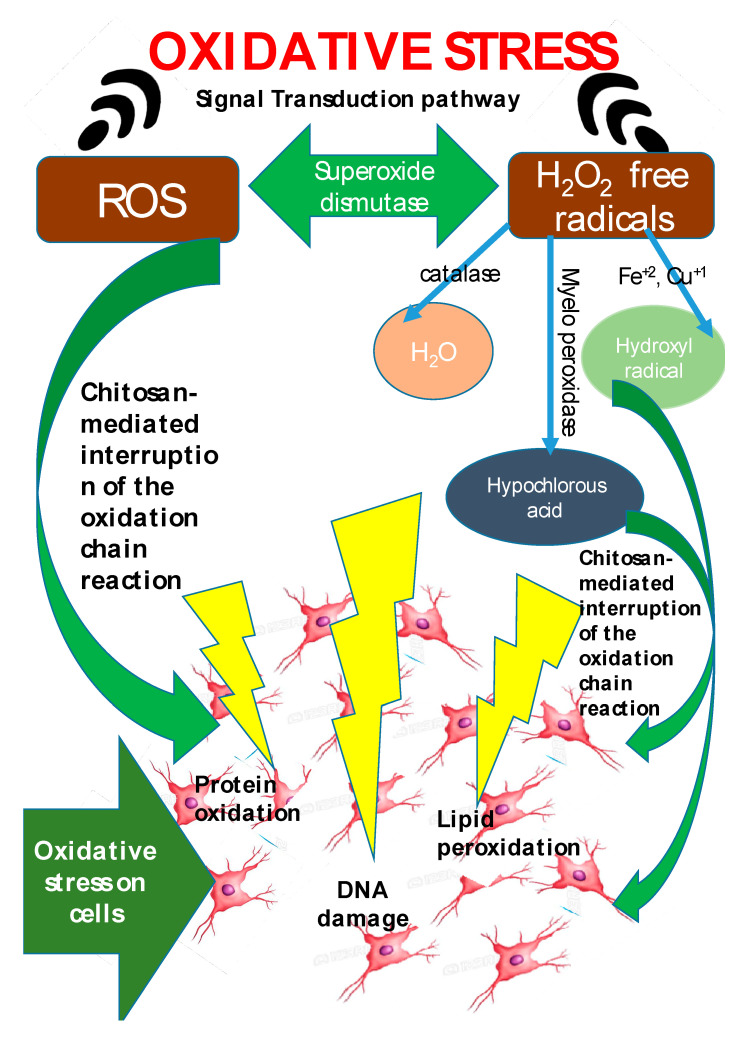
Schematic diagram showing the mode of action of antioxidant activity of chitosan via interruption of free radical chain reaction.

**Table 1 antioxidants-10-00228-t001:** Chitosan recovery methods currently in use and the varying conditions employed for deacetylation.

Source	Method	Type of Conditions	Deacetylation Degree (%)	References
Squid pens (*Loligo formosana*)	Chemical	Chitin was treated with 60% NaOH, 1/10 (*w/v*) for 60 min at 100 °C.	90%	Methacanon et al. [51]
Squid pens (*Loligo lessoniana* and *Loligo formosana*)	Chemical	Chitin was treated with 50% NaOH, 1/15 (*w/v*) for 8 h at 100 °C.	92–93%	Chandumpai et al. [52]
Lobster	Chemical	Chitin was treated with 45% NaOH, at 130 °C for 30 min.	86%	Fernandez Cerveraa et al. [53]
	Chemical	Chitin was treated with 49% NaOH, at 130 °C for 30 min.	89%	
Dublin Bay prawn shell waste (*Nephrops norvegicus*)	Chemical	Chitin (10 g) was added to reaction flasks, 100 cm^3^ of 500 g kg^−1^ NaOH solution was added to each flask, heated for 2 h at 100 °C under a nitrogen atmosphere.	81.92%	Beaney et al. [54]
Shrimp biowaste (heads or shells)	Chemical	Chitin was added with 50% NaOH, 1:20 (*w/v*) at 50 °C for 48 h.	81–84%	Rao and Stevens [55]
Shrimp shells (*Metapenaeopsis dobsoni*)	Chemical	Chitin was boiled with 40% NaOH until it got deacetylated to chitosan.	81%	Sini et al. [56]
Shrimp waste	Chemical	Chitin was added with 45% NaOH solution stirring for 90 min and heating at 130 °C.	90%	Weska et al. [57]
Crabs shells	Autoclave	Steeping in strong NaOH for 24 h before heating. Chitin was treated with 40% NaOH solution autoclaved (at 2-atmosphere pressure) for 2.5–3.0 h.	95%	Abdou et al. [36]
Crayfish shells (*Procambarus clarkia*)			95%	
Brown shrimp shells (*Penaeus aztecus*)			95%	
Pink shrimp shells (*Penaeus durarum*)			92%	
Grooved tiger prawn (*Penaeus semisulcatus*) Jinga Shrimp (*Metapenaeus affinis*) Blue Swimming Crab-Male (*Portunus pelagicus*) Blue Swimming Crab-Female (*Portunus pelagicus*) Scyllarid Lobster (*Thenus orientalis*) Cuttlefish (*Sepia* spp.)	Chemical	Chitin was treated with 45% NaOH, 1/15 (*w/v*) for 10 h at 110 °C.	88–94.4%	Sagheer et al. [43]
	Microwave	Chitin was added with 45% NaOH solution and irradiated by microwave for 15 min at 600 W.	87.5–93.0%	
Snow crab (*Chionoecetes opilio*)	Chemical	Chitin was treated with 40% NaOH solution at 105 °C for 120 min.	93.3%	Yen et al. [58]
Shrimp shells (*Metapeneaus monoceros*)	Chemical	Chitin was treated with 50% NaOH at 80 °C for 4 h.	-	Manni et al. [59]
Shrimp shells Crab shells	Chemical	Chitin (10 g) was treated with 50% NaOH at 60 °C for 8 h.	79.80% 65.89%	Zvezdova [60]
Shrimp shells (*Metapenaeus stebbingi*)	Chemical	Chitin was treated with 50% NaOH for 6 h at 120 °C.	92.19%	Kucukgulmez et al. [61]
Shrimp shells (*Parapenaeus longirostris*)	Chemical	Chitin was added in 50% NaOH solution, stirred for 3–5 h at 90–100 °C.	80%	Benhabiles et al. [62]
Crab shells (*Podophthalmus vigil*)	Chemical	Chitin was treated in 40% NaOH for 6 h at 110 °C constant stirring.	-	Prabu and Natarajan [63]
Cuttlefish (*Sepia aculeata*)	Chemical	Chitin was treated in 40% NaOH solution by heating under reflux for 6 h at 110˚C.	49.9%	Vino et al. [64]
Shrimp shells (*Metapenaeus monoceros*)	Chemical	Chitin was treated with 12.5 M NaOH, 1/10 (*w/v*) at 140 °C for 4 h.	78%	Younesa et al. [65]
Prawn shells (*Litopenaeus vannamei*)	Chemical	Chitin was treated with 50% NaOH, 1/5 (*w/v*) for 2 h at 100 °C.	80%	Mohammed et al. [66]
Fish scales (*Labeo rohita*)	Chemical	Chitin was added to 40% NaOH, 1/15 (*w/v*), and refluxed under nitrogen atmosphere for 8 h at 100 °C.	78.2%	Muslim et al. [67]
Shrimp waste	Microwave	Chitin was treated with 50% NaOH solution and irradiated by microwave for 10 min at 1400 W.	95.19%	Samar et al. [44]
Shrimp shells (*Parapenaeus longirostris*)	Chemical	Chitin was treated with 50% NaOH, 1/60 (*w/v*) for 5 h at 110 °C.	90%	Dahmane et al. [68]
Crab shells (*Carcinus mediterraneus*)	Chemical	Chitin was treated with 12.5 M NaOH, 1:10 (*w/v*) for 4 h at 140 °C.	83%	Hajji et al. [69]
Cuttlefish bones (*Sepia officinalis*)			95%	
Shrimp waste (*Penaeus kerathurus*)			88%	
Shrimp shell waste	Chemical	Chitin was added to 70% NaOH, 1/14 (*w/v*), and kept room temperature (RT) for 72 h.	74.82%	Mohanasrinivasan et al. [70]
Squid chitin	Chemical	Chitin was treated with 60% NaOH, 1/10 (*w/v*) for 60 min at 100 °C.	97.3%	Nwe et al. [71]
Crab chitin	Chemical	Chitin was treated with 40% NaOH, 1/30 (*w/v*) for 120 min at 105 °C.	93.3%	
Shrimp chitin	Chemical	Chitin was treated with 50% NaOH for 20 h at 65 °C.	87%	
Shrimp shells (*Metapenaeus monoceros*)	Chemical	Chitin was treated with 12.5 M NaOH, 1:10 (*w/v*) for 4 h at 140 °C.	81%	Younes et al. [72]
Shrimp shells	Chemical	Chitin was refluxed with 50% NaOH, 1/10 (*w/v*) at 90 °C for 4 h.	95.5%	Abdel-Rahman et al. [73]
Crab shells (*Carcinus mediterraneus*)	Chemical	Chitin was treated with 12.5 M NaOH, 1/10 (*w/v*) for 4 h at 140 °C	83%	Hajji et al. [74]
Cuttlefish bones (*Sepia officinalis*)			95%	
Shrimp waste (*Penaeus kerathurus*)			88%	
Fish scales (*Labeo rohita*)	Chemical	Steeping in strong NaOH for 24 h before heating. Chitin was treated with 40–50% NaOH for 5–6 h at 100–160 °C.	61%	Kumari et al. [75]
Shrimp shell waste	Chemical	Chitin was treated with 48% NaOH for 48 h at 25 °C.	70–85%.	Ahing and Wid [76]
Shrimp waste	Autoclave	Chitin was added in 50% NaOH, 1/10 (*w/v*), and autoclaved for 20 min at 15 psi/121 °C.	70.9%	Al-Hassan [37]
Spawning of veined rapa whelk (*Rapana venosa*)	Chemical	Ten grams of the sample was soaked in 4% NaOH, 1/15 (*w/v*) at 65 °C for 2 h.	-	Apetroaei et al. [77]
Warty crab shells (*Eriphia verrucosa*)	Chemical	Chitin was treated with 45% NaOH, 1:20 (*w/v*) at 100 °C, for 15 min.	-	
Fish scales (*Oreochromis niloticus*)	Chemical	Chitin was added in 40% NaOH, 1/40 (*w/v*), stirred for 6 h (300 rpm) at 117 °C.	97.5%	Boarin-Alcalde and Graciano-Fonseca [78]
Blue crab shell wastes (*Callinectes sapidus*)	Chemical	Chitin was added in 50% NaOH, 1/10 (*w/v*), and heated for 4 h at 150 °C.	85%	Demir et al. [79]
Crayfish shells	Chemical	Chitin was treated with 60% NaOH at 100 °C for 4 h.	(Pure chitosan was obtained)	Duman and Kaya [80]
Shrimp shells (*Parapenaeus longirostris*)	Chemical	Chitin was deacetylated with 15 M NaOH, 1/20 (*w/v*) at 110 °C under vacuum and constant stirring for 5 h.	73.68%	Hafsa et al. [47]
	Ultrasonic	Chitin was suspended with 15 M NaOH, 1/20 (*w/v*), irradiated by ultrasonic (v = 50 kHz) for 3 h.	83.55%	
Squid pens (*Loligo japonica*)	Chemical	Chitin was treated with 40% NaOH at 95 °C for 6 h.	91.04%	He et al. [81]
Cuttlebone (*Sepia pharaonic*)	Chemical	Chitin was treated in hot concentrated NaOH (40–50%) solution to yield chitosan, which was sulfated using chlorosulfonic acid and stirred for 30 min to obtain sulfated chitosan.	81%	Karthik et al. [82]
Shrimp shell (*Crangon crangon*)	Chemical	Chitin was refluxed in NaOH (50% by weight) at 90 to 100 °C, stirred for 6 h.	76%	Kumari et al. [83]
Fish scale (Labeorohita)			80%	
Norway lobster (*Nephrops norvegicus*)	Chemical	Chitin was treated with 50% NaOH for 4 h at 120 °C.	71.59%	Sayari et al. [84]
Shrimp shell (*Litopenaeus vannamei*)	Autoclave	Extracted with hot sulfuric acid at 95 °C overnight and autoclaved for 20 min at 121 °C.	76%	Vilar Junior et al. [38]
	Chemical	Treated with 40–50% NaOH, 1/20 (*w/v*) at 100–120 °C for 60–720 min.	81.7%	
Squid gladius (*Sepioteuthis* *lessoniana*)	Chemical	Chitin was treated with 50% NaOH at 120 °C for 4 h.	71%	Abdelmalek et al. [85]
Shrimp shells (*Parapenaeus longirostris*)	Chemical	Chitin was treated with 28.6% NaOH at 81.15 °C for 9.55 h.	98%	Ben Seghir et al. [86]
Crab shells (*Crangon crangon*)	Chemical	Chitin was treated with 40% KOH for 6 h at 90 °C.	70%	Kumari et al. [87]
Fishery waste (*Labeo rohita*)			75%	
Shrimp shells (*Crangon crangon*)			78%	
Shrimp shells Crab shells	Chemical	Chitin was treated with 65% NaOH for 72 at 30 °C.	88.48% 80.12%	Premasudha et al. [88]
Shrimp shells (*Litopenaeus vannamei*)	Chemical	Chitin was added with 12.5 M NaOH, 1:15 (*w/v*) cooled, kept at −83 °C for 24 h, and stirring (250 rpm) for 4 or 6 h at 115 °C.	91%	de Queiroz Antonino et al. [89]
Squid pin (*Doryteuthis singhalensis*)	Chemical	Chitin was treated in 40% NaOH solution by heating under reflux for 6 h at 110˚C.	83.76%	Ramasamy et al. [90]
Shrimp waste (*Penaeus merguiensis*)	Autoclave	Chitin was treated with 45% NaOH, 1/15 (*w/v*), and autoclaved for 30 min at 15 psi/121 °C.	88%	Sedaghat et al. [39]
	Microwave	Chitin was treated with 50% NaOH and irradiated by microwave for 10 min at 1400 W.		
	Chemical	Chitin was treated with 50% NaOH at a 1/5 (*w/v*) ratio for 2 h at 100 °C.		
Shrimp shell waste	Autoclave	One gram of chitin was added in 50% NaOH and autoclaved for 1 h at 121 °C, 15 psi.	-	Varun et al. [40]
Fish scales (Red Snapper)	Chemical	Chitin was treated with 80% NaOH, 1/3 (*w/v*) for 4 h at 110 °C.	90.83%	Takarina and Fanani [91]
Fish scales (White Snapper)	Chemical	Chitin was treated with 80% NaOH for 4 h at 120 °C.	84.05%	Takarina et al. [92]
Blue crab shells (*Portunus segnis*)	Chemical	Chitin was treated with 12.5 M NaOH, 1/10 (*w/v*) for 4 h at 140 °C.	90.39%	Hamdi et al. [93]
Squid pen (*Illex argentines*)	Chemical	Chitin was dissolved in 5% acetic acid, filtered, and precipitated with 8 N NaOH solution, washed with reverse osmosis (RO) water until pH reached 7.0.	84%	Huang et al. [94]
Prawn shells	Chemical	Chitin was refluxed in 50% NaOH solution for 30–150 min at 100 °C.	78.40%	Muley et al. [95]
Crab shells (*Portunus sanguinolentus*)	Chemical	Chitin was treated with 80% NaOH, 1:20 (*w/v*) at 90–95 °C for 5h.	70.79%	Rubini et al. [96]
Shrimp shells (*Penaeus monodon*)	Chemical	One gram of chitin in 50 mL of 50% NaOH, stirred for 50 min at 90 °C, filtered, and treated with 80% alcohol in 1/30 (*w/v*) ratio for 24 h at 80 °C.	65%	Srinivasan et al. [97]
Shrimp waste	Chemical	Chitosan-1: 40% NaOH 1/20 (*w/v*) at 120 °C for 300 min. Chitosan-2: 50% NaOH 1/20 (*w/v*) at 100 °C for 720 min.	78.2% 84.95%	Tokatli and Demirdöven [98]
Lobster shells (*Thenus unimaculatus*)	Chemical	Chitin was added with 40% NaOH, stirred for 6 h at 110 °C, filtered, treated with 10% acetic acid for 12 h.	-	Arasukumar et al. [99]
Shrimp shells waste	Chemical	Chitin was treated with 50% NaOH under agitation for 4 h at 90 °C.	88%	Boudouaia et al. [100]
Shrimp shells (*Parapenaeus longirostris*)	Microwave	Chitin was treated with 40% NaOH, 1:20 (*w/v*), heated by microwave at 650 W for 12 min.	82.8%	EL Knidri et al. [45]
Blue crab shell (*Callinectes sapidus*)	Chemical	Chitin was treated with 50% NaOH, 1/10 (*w/v*) at 150 °C for 4 h.	71%	Metin et al. [101]
Shrimp shell	Chemical	Chitin was treated with 50% NaOH at 60 °C, dry residue was added into 2% (*w*/*w*) acetic acid, 30% H_2_O_2_ was added and kept for 4 h.	64.18%	Ni’mah et al. [102]
Mussel shell			35.03%	
Squid pen			58.04%	
Crab shell			53.91%	
Crab shell waste	Chemical	Chitin was treated with 50% NaOH, 1/10 (*w*/*v*) for 100 min at 100 °C.	82%	Pădurețu et al. [103]
Shrimp waste	Chemical	Chitin was added with 50% NaOH, 1/15 (*w/v*), stirred for 2 h (1 h at RT and 1 h at 100 °C).	84%	Pădurețu et al. [104]
Squid pens (*Loligo formosana*)	Chemical	Chitin was treated with 50% NaOH, 1:50 (*w/v*) for 8 h at 130 ℃.	90%	Singh et al. [105]
Shrimp waste (*Litopenaeus vannamei*)	Microwave 1	Chitin (16, 32, 60 mesh sizes) was treated 45% NaOH, 1/15 (*w/v*) irradiated in 6 pulses of 5 min at 600 W.	81, 72, 78%	Santos et al. [46]
	Microwave 2	Between each interval, they were stirred for homogenization.	81, 92, 89%	
Shrimp shells	Chemical	Chitins were mixed with 40% NaOH, 1:10 (*w/v*), stirred overnight, and the mixture was heated for 12 h at 100 °C.	93%	Tolesa et al. [106]
Fish waste (*Sardina pilchardus*)	Autoclave	Chitin was added with 40% NaOH and autoclaved for 20 min at 15 psi/121 °C.	87%	Aboudamia et al. [41]
Shrimp shell waste	Chemical	Chitin was added with 50% NaOH, stirred for 1 h at 30 ℃.	88.89%	Aldila et al. [107]
Shrimp residues (*Farfantepenaeus aztecus*)	Ultrasound	Two grams of shrimp residues powder in 50 mL of CaCl_2_–MeOH–H_2_O, stirred for 20 min at 60 °C, ultrasound at 40 kHz 30 min at 60 °C, rest for 48 h at RT.	65.87%	Borja-Urzola et al. [48]
Crab shell (*Portunus trituberculatus*)	Ultrasonication	Chitin was treated with 50% NaOH, 1:15 (*w/v*), at 75 ℃ for 3.5 h with sonication.	86.02%	Huang et al. [49]
Omani shrimp waste	Autoclave	Chitin was treated with 50% NaOH, 1:10 (*w/v*), autoclaved for 15 min at 121 °C.	-	Said Al Hoqani et al. [42]
Shrimp shells (*Litopenaeus vannamei*)	Chemical	Chitin samples (0–9) were treated with 12.5 M NaOH, 1:5 (*w*/*v*) at 65 °C for 12 h.	56.10–88.76%	Trung et al. [108]
Shrimp shell and Crab Shells	Graviola extract with magnetic stirring	Shells interacted with graviola leaf extract with magnetic stirring.	50.97–94.56%	Gopal et al. [50]

**Table 2 antioxidants-10-00228-t002:** Various nano-chitosan morphologies and their medical applications.

Nano-chitosan Morphology	Preparation Method	Applications	References
Nanogels	Covalent cross-linking, ion crosslinking, covalent modification	Photothermal therapy–chemotheraphy, controlled drug delivery, deep tumor penetration	[245,246,247,248]
Micelles	Covalent modification/self assembly, ion crosslinking	Drug delivery, photodynamic theraphy, ocular delivery	[249,250,251,252,253]
Nanofibers	Electrospinning process	Improve osteogenic activity	[254]
Liposomes	Covalent modification/self assembly, electrostatic adsorption	Anticancer drug delivery, antimalarial drug delivery, reverse drug resistance, photothermal, and chemotheraphy	[255,256,257,258,259]
Nanosphere	Covalent modification/self assembly, electrostatic adsorption, emulsification	siRNA delivery, drug delivery, drug release	[260,261,262]
Nanoparticles	Covalent modification/self assembly, ion crosslinking	Oral delivery, siRNA delivery, targeted therapy	[263,264,265,266]

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
