# Peer review of "Crustacean Waste-Derived Chitosan: Antioxidant Properties and Future Perspective"

_antioxidants, 2021, doi:10.3390/antiox10020228_

Round 1

Reviewer 1 Report

General comments/questions:

- I have some concerns regarding the statement “chitosan extraction” (see Table 1). In my opinion, this statement rather refers to the recovery of chitosan being already present in the material than, the chitosan obtained in a deacetylation process of previously “extracted chitin”

- what is the role of Table 1 in this Review – it constitutes a simple collection of data commonly found in the literature, especially as in general chitin deacetylation process is performed through alkaline treatment of chitin. There is a lack of discussion of these data. What Authors want to show or on what Authors want to put their attention?

- I expected more information about the chitosan antioxidant properties. Are there any literature reports describing the possible mechanism of antioxidative chitosan acting?

Particular comments:

- line 42 – please use more recent data (literature cited [6,7] was published in 1991 and 2003)

- line 52 – not only solubilization but in case of CaCO3 also decomposition

- line 60 – change “its’ into “chitin”

- line 64 - in my opinion not only HCl is used in the demineralization process, however, HCl is the most commonly used

- line 107 – NH2

- line 142 – in the case of chitosan it is better to use “molecular weight” rather than “degree of polymerization”

- line 314 – lit number should be added

Author Response

We would like to thank the Editor and the reviewers team for their meticulous efforts on our manuscript. We are so very grateful for the opportunity to revise our manuscript. We humbly accept all the valuable suggestions. We have revised the manuscript as per your suggestions as much as was possible. We have highlighted the changes to the manuscript using track changes. We present below a point by point response to the queries raised. We thank you very much for your time and inputs. Thank you.

Reviewer 1

General comments/questions:

- I have some concerns regarding the statement “chitosan extraction” (see Table 1). In my opinion, this statement rather refers to the recovery of chitosan being already present in the material than, the chitosan obtained in a deacetylation process of previously “extracted chitin”

Ans. Yes, we absolutely agree with you, we have now replaced this term with chitosan recovery in the table and also wherever applicable in the text. We thank you for rightly pointing out.

- what is the role of Table 1 in this Review – it constitutes a simple collection of data commonly found in the literature, especially as in general chitin deacetylation process is performed through alkaline treatment of chitin. There is a lack of discussion of these data. What Authors want to show or on what Authors want to put their attention?

Ans. There are ennumerable tables on the extraction of chitin available across many publications. A handful tables have been published on chitosan recovery basically highlighting just the broad classifications of alkaline chemical extract and biological extraction. We have presented in Table 1 an upto date comprehensive summary of all the recovery methods reported specifically with respect to recovery of chitosan from marine shell wastes and the techniques involved with respect to chitosan recovery. We do accept that maybe our lack of a discussion on table 1 could have been the reason why you felt that there was a lacuna. We have now discussed the data in the revision. Thank you.

- I expected more information about the chitosan antioxidant properties. Are there any literature reports describing the possible mechanism of antioxidative chitosan acting?

Ans. The reason why there is less information in the manuscript on the antioxidant activity of chitosan is because, this property has been really less highlighted in published research. We have once again revisited the available resources and added on as much as possible, we have also highlighted the inference that this area has not been well documented. As indicated in literature, a clear cut mechanism is not yet available for the antioxidant activity of chitosan, however, we have drawn a plausible schematic representation (based on Riaz, Muhammad Shahid & Zhao, Liqing & Mehwish, Mahreen & Wu, Yiguang & Mahmood, Shahid. (2019). Chitosan and its derivatives: synthesis, biotechnological applications, and future challenges. Applied Microbiology and Biotechnology. 103. 10.1007/s00253-018-9550-z. a) of the mechanism in Figure 2. We have provided a brief description based on the available information in the revised text. Thank you. 

Particular comments:

- line 42 – please use more recent data (literature cited [6,7] was published in 1991 and 2003)

Ans. Added recent data. Thank you.

- line 52 – not only solubilization but in case of CaCO3 also decomposition

Ans. Yes changed. Thank you.

- line 60 – change “its’ into “chitin”

Ans. Changed.

- line 64 - in my opinion not only HCl is used in the demineralization process, however, HCl is the most commonly used

Ans. Yes, rephrased.

- line 107 – NH2

Ans. Sorry, changed.

- line 142 – in the case of chitosan it is better to use “molecular weight” rather than “degree of polymerization”

Ans. Yes, rephrased.

- line 314 – lit number should be added

Ans. Added. Thank you.

Reviewer 2 Report

This is a good review paper that gives an overview of arthropod- derived chitosan, including many interesting extraction techniques and antioxidant properties. Bioextraction is presented as a promising technique that has been heavily researched on a laboratory scale. The article is timely and appropriate to the journal, but the following improvements are needed:

1.Many similar review articles have already been published. Please justify the originality of your review (what is new) in the introduction.

2.Authors should put more effort to improve the organization and readability. Here, some figures and additional tables will help. For example, Table 1 is quite informative with many chitosan extraction methods listed. Therefore, I strongly suggest to include additional figures or tables on chitin and chitosan interesting biological properties with focus on bioactive and antioxidant properties and nano aspects.

Author Response

This is a good review paper that gives an overview of arthropod- derived chitosan, including many interesting extraction techniques and antioxidant properties. Bioextraction is presented as a promising technique that has been heavily researched on a laboratory scale. The article is timely and appropriate to the journal, but the following improvements are needed:

Ans. Thankyou for the encouraging words. We have made the improvements you have suggested. Thank you again. 

1.Many similar review articles have already been published. Please justify the originality of your review (what is new) in the introduction.

Ans. We understand your concern, we have now justified the uniqueness of this article in the introduction. Thank you.

2.Authors should put more effort to improve the organization and readability. Here, some figures and additional tables will help. For example, Table 1 is quite informative with many chitosan extraction methods listed. Therefore, I strongly suggest to include additional figures or tables on chitin and chitosan interesting biological properties with focus on bioactive and antioxidant properties and nano aspects.

Ans. Thank you for the suggestion, we have now added a table as Table 2 and a Figure as Figure 2 in the revised version submitted. We thank you for prompting us.

Reviewer 3 Report

The manuscript Crustacean waste-derived chitosan: antioxidant properties and  future perspective  submitted to Antioxidants describes the recovery of chitosan from crustacean wastes and also the physicochemical and biological properties of chitosan. The article is original and well structured, and presents future prospects. I do not find important mistakes. Consequently, I recommend the manuscript for publication.

Author Response

We would like to thank the Editor and the reviewers team for their meticulous efforts on our manuscript. We are so very grateful for the opportunity to revise our manuscript. We humbly accept all the valuable suggestions. We have revised the manuscript as per your suggestions as much as was possible. We have highlighted the changes to the manuscript using track changes. We present below a point by point response to the queries raised. We thank you very much for your time and inputs. Thank you.

The manuscript Crustacean waste-derived chitosan: antioxidant properties and  future perspective  submitted to Antioxidants describes the recovery of chitosan from crustacean wastes and also the physicochemical and biological properties of chitosan. The article is original and well structured, and presents future prospects. I do not find important mistakes. Consequently, I recommend the manuscript for publication.

Ans. We are very grateful for the words of appreciation. Thank you very much. Greatly encouraged.

Round 2

Reviewer 1 Report

  • in my opinion, the caption of Figure 2 should be changed - as it does not refer to the chemical mechanism, rather to the "action"
  • line 309:  change "." into ","
  • the title should be changed as also production in the deacetylation process are widely described (table 1)

Author Response

  • in my opinion, the caption of Figure 2 should be changed - as it does not refer to the chemical mechanism, rather to the "action"
  • line 309:  change "." into ","
  • the title should be changed as also production in the deacetylation process are widely described (table 1

Ans. Thankyou, we have indeed changed the caption for figure 2

line 309----changed

and table 1 title also we have changed. thank you for ur interest and time on our manuscript. 

Reviewer 2 Report

The paper was improved from the initially submitted version. Clarity of the results presented was improved due to the insertion of additional figures and editing of originals. The limitations of the study have been more explicitly stated by the authors.

Author Response

The paper was improved from the initially submitted version. Clarity of the results presented was improved due to the insertion of additional figures and editing of originals. The limitations of the study have been more explicitly stated by the authors.

Ans

thankyou for your acknowledgment of our revisions. thankyou for your valuable inputs. thank you again